# Comprehensive evaluation of patterns of hypoglycemia unawareness (HUA) and glycemic variability (GV) in patients with fibrocalculous pancreatic diabetes (FCPD): A cross-sectional study from South India

Riddhi Dasgupta[1], Felix K. Jebasingh[1], Shajith Anoop[1], Santhya Seenivasan[1], Mathews Edatharayil Kurian[1], Flory Christina[1], Gracy Varghese[2], Pamela Christudoss[2], K. U. Lijesh[1], Deepu David[3], Sudipta Dhar Chowdhury[3], Thomas V. Paul[1], Nihal Thomas[1] *

1 Department of Endocrinology, Diabetes and Metabolism, Christian Medical College, Vellore, India,
2 Department of Biochemistry, Christian Medical College, Vellore, India, 3 Department of Gastroenterology, Christian Medical College, Vellore, India

☯ These authors contributed equally to this work.
* nihal_thomas@cmcvellore.ac.in

## Abstract

### Objectives

Hypoglycemia unawareness (HUA) in patients with FCPD is common with an unclear etiology. We evaluated the prevalence, characteristics of HUA, glycemic variability (GV), its possible association with pancreatic glucagon secretion & cardiac autonomic function in patients with FCPD.

### Methods

A two-week ambulatory glucose profile (AGP) and cardiac autonomic function test was done in patients with FCPD (n = 60), and categorized into UNAWARE (n = 44) and AWARE (n = 16) groups based on the Hypoglycemia Unawareness Index (HUI) score. Glycaemic variability was assessed from the AGP data using Easy GV 9.0.2 software. A subset of patients from both the groups (n = 11) underwent a mixed-meal challenge test and were compared with healthy individuals (controls; n = 11).

### Results

HUA was evidenced in 73% (44/60) of patients with FCPD. Significant hypoglycemia, nocturnal hypoglycemia, duration of hypoglycemia and poor cardiac autonomic functions (p = 0.01) were prominent in the UNAWARE group. The overall GV was greater in the UNAWARE group. In the UNAWARE group, significantly reduced fasting and post prandial glucagon levels negatively correlated with HUI ($r$ = -0.74, p < 0.05) and GV-hypoglycemia

**Data Availability Statement:** All relevant data are available in the paper and in the Supporting Information files.

**Funding:** This study was supported by a research grant from the Research Society for Study of Diabetes in India (RSSDI); Grant No: RSSDI/HQ/ GRANTS/ 2018/460. The funding body had no role in study design, data collection and analysis, preparation, review of the manuscript, selection of journal and decision to publish.

**Competing interests:** The authors have declared that no competing interests exist

indices (p < 0.05) In contrast, significantly higher post prandial glucagon levels in the AWARE group positively correlated with post prandial hyperglycemia ($r = 0.61$, p < 0.05).

## Conclusion

Heterogeneity in patterns of glucagon secretion were significantly associated with HUA and GV. Reduced glucagon levels contribute to greater risks of HUA, nocturnal hypoglycemia and greater GV, while hyperglucagonemia predisposes to postprandial hyperglycemia and hypoglycemia awareness in patients with FCPD.

## Introduction

The global prevalence of pancreatic diabetes is less than 1%. However, its prevalence is higher in tropical countries [1] with studies reporting a prevalence of up to 4% [2–4]. Fibrocalculous Pancreatic Diabetes (FCPD) contributes a significant proportion of diabetes burden in tropical countries such as India [4], whereas the most common etiology of pancreatic diabetes in Western countries is alcohol- induced pancreatitis and its sequelae [5].

FCPD is an important clinical entity featured by progressive inflammation and fibrosis of the pancreas, leading to exocrine and endocrine insufficiency, fat mal-absorption and overt hyperglycemia. Various mechanisms had been postulated for hyperglycemia in patients with FCPD, which include beta cell loss [5, 6], insulin secretory dysfunction [7, 8], receptor and signal transduction defects leading to insulin resistance [9, 10], and reduced incretin effect [11, 12], with the latter being linked to pancreatic exocrine insufficiency [13]. Patients with FCPD experience wide fluctuations in blood glucose ranging between hypoglycemia and hyperglycemia, making such patients predisposed to hypoglycemia unawareness (HUA) and high glycemic variability (GV) [14]. This could be attributed to the lack of glucagon secretion, due to extensive pancreatic destruction [15]. Certain studies in patients with FCPD have reported preserved or even exaggerated post prandial glucagon secretion at one hour in response to an oral glucose tolerance test [16]. Studies in de-pancreatomised animal models have shown similar results, suggestive of an extra-pancreatic source of glucagon during acute insulin deficiency [17].

Glycemic variability (GV) is an independent risk factor for the development of micro and macro vascular complications of diabetes mellitus [18, 19], Capillary blood glucose readings often fail to represent the glycemic fluctuations with accuracy, including episodes of nocturnal hypoglycemia, whereas HbA1C provides an integrated measurement of glycemic trends over a sustained period of time but fails to address GV parameters [20, 21]. In this scenario, continuous glucose monitoring (CGM) would be an ideal alternative as it provides adequate data on daily glycemic excursions and nocturnal glucose levels [22]. However, there is a paucity of literature utilizing CGM-based metrics to evaluate the patterns of hypoglycemia and GV in patients with FCPD.

Previous studies have shown the prevalence of autonomic dysfunction in patients with diabetes [23]. Chronic, uncontrolled diabetes leads to autonomic dysfunction, which in turn attenuates hypoglycemia awareness, especially in patients with FCPD. We hypothesized that glucagon levels may be associated with hypoglycemia unawareness (HUA) and cardiac autonomic dysfunction in patients with FCPD. Therefore, we aimed to objectively assess the prevalence and patterns of GV and HUA in patients with FCPD and their possible association with

glucagon levels. The secondary objective was to compare the characteristics of cardiac autonomic dysfunction in patients with FCPD presenting with or without HUA.

## Methods

### Study design

This cross-sectional study was approved by the institutional review board (IRB) for ethics in research on humans (IRB number 10788 dated 01 August 2017) of Christian Medical College (CMC) Vellore India. The study was conducted in patients who presented with FCPD over a period of one year (Between August 2017 and July 2018). FCPD is characterized as a combination of recurrent pancreatitis with abdominal pain, pancreatic ductal calculi or ductal dilatation on imaging, pancreatic exocrine insufficiency and diabetes mellitus in the absence of alcoholism [24]. Adults aged 18 to 45 years with clinically diagnosed FCPD for a minimum duration of 12 months and Hba1c between 7–10% were included in the study with informed consent. The study adhered to guidelines of the declaration of Helsinki, 2013.

### Sample size calculation

With a prevalence rate of FCPD in India being 0.2% and the incidence of hypoglycemia ranging between 40–45% in such patients, a sample size of 60 subjects was calculated using the formula $= \frac{Z^2 \text{ x P x } (1-P)}{d^2}$, where P denotes the prevalence rate at 95% confidence interval (Z = 1.96) and margin of error (d) being ± 10%. A total of 110 patients with FCPD were screened by homogenous, purposive sampling method and 60 patients with FCPD who fulfilled the eligibility criteria were included in the study. An additional group of healthy, non-diabetic, age, sex and BMI-matched subjects (n = 11) were recruited as controls with informed consent. (**Fig 1: Algorithm of the study.**).

### Exclusion criteria

Patients with chronic pancreatitis other than FCPD were excluded. The exclusion criteria included patients with FCPD and comorbidities namely, chronic kidney disease stage 4 and 5, chronic alcoholism, chronic liver disease, coronary artery disease, epilepsy or unexplained blackouts, untreated hypothyroidism, longstanding hypo-adrenalism, evidence of pancreatic carcinoma, bleeding disorders, patients on statin therapy, or beta blockers and pregnant or lactating women. Those patients with poor understanding of the disease or inability to understand the use of Flash Glucose Monitoring System (FGMS) were also excluded from the study.

### Screening and recruitment

The initial screening for hypoglycemia patients with FCPD was done using the following parameters:

a. Clarke's questionnaire [25] for hypoglycemia with a cut-off score of more than 4

b. Clinical history of ≥ 2 episodes of severe hypoglycemia the last 12 months

c. Clinical history of ≥ 4 episodes of hypoglycemia on capillary blood glucose readings in the last 4 weeks.

The Clarke's questionnaire is an 8-item questionnaire used for the subjective assessment of awareness of hypoglycemia in patients with FCPD (**S1 File**). Severe hypoglycemia was defined as episodes of low blood glucose levels requiring hospitalization or medical assistance. All the study participants were asked to do a 6-point (pre and 2-hour post meal daily) glucose profile

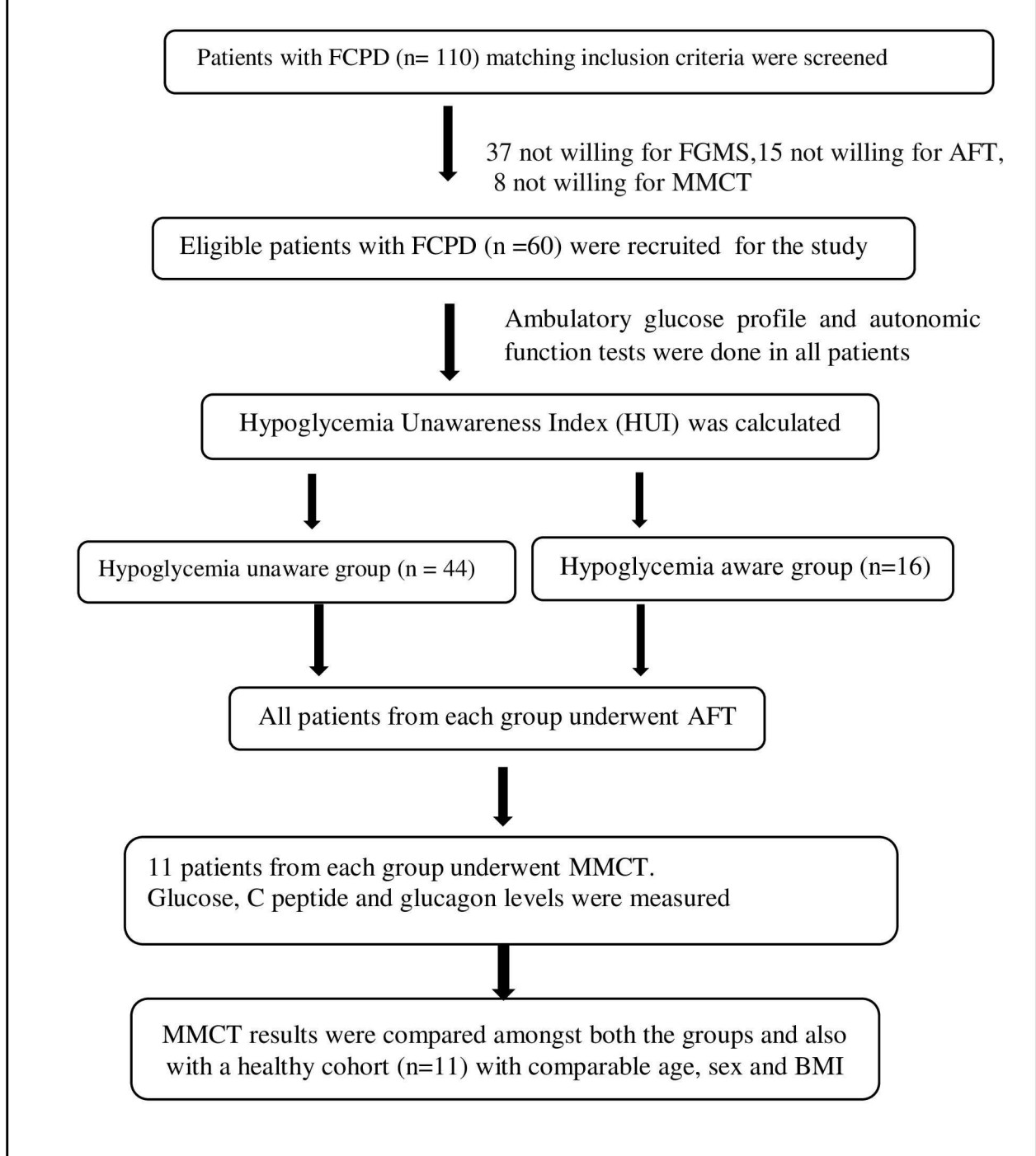

FCPD : Fibrocalculous Pancreatic Diabetes, FGMS : Flash Glucose monitoring system.
MMCT: Mixed meal challenge test, AFT: Autonomic function testing, BMI: Body mass index.

**Fig 1. Algorithm of the study.**

monitoring using a glucometer for 4 weeks and document it in a diary provided by the clinician. The glucometer data was assessed for the frequency of episodes of hypoglycemia.

Hypoglycemia was defined as per the American Diabetes Association (ADA) diagnostic criteria: capillary glucose levels < 70mg/dl. Clinical hypoglycemia was defined as capillary blood glucose levels ≤ 54mg/dl and severe hypoglycemia was defined as any hypoglycemic episodes requiring assistance/ hospitalization [26]. Those patients who fulfilled at least one of the other two criteria were included in the study. The participants (n = 60) underwent ambulatory glucose profile (AGP) monitoring for 14 days using a Flash Glucose Monitoring Device (FGMS) sensor (Free style Libre Pro, UK®) and cardiac autonomic function testing on the Cardiac autonomic function system analyser (CANS 504®, Diabetik Foot Care Pvt Ltd, India).

## Hypoglycemia unawareness index (HUI) and patterns of hypoglycemia

The Free style Libre Pro® FGMS sensor was affixed securely on the upper aspect of the non-dominant arm of participants to automatically assesses interstitial blood glucose level every 15 minutes, and record data up to 14 days. The FGMS sensor was retained in functional mode for 14 days and the data was downloaded using the FGMS device on a daily basis. Patterns of hypoglycemia were noted from the AGP data of the FGMS device. Patients were instructed to record a 6-point capillary glucose profile including 3.00 am capillary blood glucose levels using a glucometer on alternative days, to corroborate with the data obtained by FGMS. The participants visited the clinic on alternate days to report any adverse glycemic excursions and to check if the sensor was in functional mode. In addition, patients were asked if they had experienced symptoms of hypoglycemia at the corresponding time of documented hypoglycemia.

The HUI was calculated using the formula mentioned below.

$$\text{Hypoglycemia Unawareness Index (HUI)} = \frac{\text{Number of episodes of hypoglycemia unawareness}}{\text{Total number of episodes of hypoglycemia unawareness}}$$

Patients with FCPD were classified into 2 groups based on HUI as hypoglycemia aware (AWARE) and hypoglycemia unaware (UNAWARE). Those patients who reported hypoglycemia unawareness index of ≥ 0.3 were categorized as AWARE, and those with hypoglycemia unawareness index < 0.3 were categorized as UNAWARE. Hypoglycemia patterns were assessed from the FGMS readings in both groups as follows:

1. The total number of hypoglycemic episodes were defined as the total number of times the interstitial blood glucose level were less than 70mg/dl during the 14 days when the FGMS sensor was affixed.

2. The number of nocturnal hypoglycemic episodes (interstitial blood glucose level less than 70 mg/dl) during nocturnal sleep between 9 pm and 6 am over the last 14 days.

3. Severe hypoglycemia was defined as any hypoglycemic episode which required assistance in the form of an intra-venous glucose infusion.

4. Clinically significant hypoglycemia was defined as an interstitial blood glucose level of less than 54 mg/dL.

## Cardiac autonomic function testing

The assessment of autonomic function was done using an automated Cardiac Autonomic Neuropathy System Analyzer (CANS 504, Diabetic Foot Care India Pvt Ltd, India ®)—a computer-based analyzer used to assess both sympathetic and parasympathetic autonomic nervous system response to an activity (Sensitivity: 86%, Specificity: 73%). The patients were instructed not to smoke or consume caffeine containing beverages or chocolates for atleast 3 hours prior

to testing. Patients with cardiac pacemaker or history of acute cardiac failure / cardiac disease were not included for this procedure.

The tests for autonomic dysfunction were performed as per the standard protocols stipulated by the manufacturer. The CANS system used an electrocardiogram and an automatic blood pressure monitoring module at resting and standing postures (R-R interval). The evaluation of parasympathetic nervous system functions included heart rate variability (RR interval ratio) between the longest RR interval and the shortest RR interval in response to deep breathing, standing and a continuous electro cardiogram (ECG) recording in response to exerted breath. The tests for sympathetic system included assessment of the maximum increase in blood pressure on sustained handgrip and a postural drop in blood pressure on standing. The details of the procedure are provided in **S2 File**.

## Mixed meal challenge test (MMCT)

A subset of patients with FCPD in the AWARE and UNAWARE groups (both; n = 11) were chosen for the MMCT. They were provided with a standard meal and snack to consume the night prior to the MMCT and were fasting after 10 pm. The participants reported to the laboratory the next day and blood samples were drawn in the fasting state. Following this, the patients were provided with a drink of "Ensure Plus®", (Abbott Health care Pvt Ltd, India)–a nutritional supplement (composed of carbohydrates 54%, fat 32% and protein 14%). For the MMCT, 6 scoops (equivalent to 219 kcal) of the meal mixture were dissolved in 250 ml water and consumed by the patient. Blood samples were drawn through an indwelling intravenous catheter at 15, 30, 45, 60, 90-, 120-, 150- and 180-minutes following meal consumption, for the measurements of glucose, glucagon and C-peptide levels. Blood samples collected in coagulant free tubes were allowed a period of 20 minutes at room temperature for clotting and separation of serum. All samples were checked to ensure no hemolysis occurred. Serum was separated by centrifugation at 2500 rpm for 15 minutes in a refrigerated centrifuge at 4 degrees centigrade.

Blood samples for glucagon assay were collected in sterile glass tubes containing Aprotinin solution (500 kIU/ml blood; Trasylol, Bayer, Leverkusen, Germany), 0.3 ml of EDTA and protease inhibitors (Complete, Mini, EDTA- free", Roche applied sciences, Germany). The sample tubes were stored on ice and were centrifuged at 2000g for 15 minutes in a refrigerated centrifuge to separate the plasma. All plasma/serum samples were stored at minus 20˚C until assay. Glucagon levels were assessed using ELISA kits (Mercodia®, USA), which has high sensitivity to detect glucagon levels (Detection limit: < 1 pmol/L, Intra-assay coefficient of variation (CV): 3.3–5.1%, inter-assay CV: 7.5%). Glucagon secretion rate was calculated from the Area-under-curve (AUC) using the trapezoidal method. C-peptide levels were measured by Chemi-luminesent immunometric assay (CLIA) using enzymatic kits (IMMULITE 2000; Intra- assay CV: 4% Inter-assay CV 7.3%). Plasma glucose levels were assessed by the hexokinase method (intra-assay CV: 0.4–1.2%, inter assay CV: 5.3%).

## Assessment of glycemic variability (GV) using FGMS data

The glycemic variability patterns in patients with FCPD were assessed using the Easy GV version 9.0.2 software, (Oxford University, Freelance Version). The FGMS data recorded over 14 days was converted to mmol/L from mg/dL and computed into the software. Complete data were analysed and the indices of GV were reported in mmol/L. This data was reconverted into mg/dl. After calculating the individual data of patients, the data from both the groups were compiled separately and the mean value of indices of the following indices of GV were calculated. Standard deviation of the mean of the sensor glucose values (SD), Coefficient of variance (% CV), Continuous Overall Net Glycemic Action (CONGA), Mean Amplitude of Glycemic

Excursion (MAGE), Absolute Means of Daily Differences (MODD)and Average Daily Risk Range (ADRR) (**S3 File**).

## Statistical analysis

Data were checked for normative distribution. For categorical variables, data was reported as actual number & (%) and for continuous variables as mean ± SD. Difference between the two groups were assessed using Chi-square and Student's *t*-test. Correlation between variables were assessed using Pearson's and Spearman's coefficients. Statistical significance was set at a *P* value of less than 0.05. Data analysis was performed using SPSS software for Windows (Version 21.0; SPSS Inc, Chicago, IL, USA).

## Results

In this study, 73% (n = 44/60) of patients with FCPD had hypoglycemia unawareness. The results for all the study parameters were compared between the UNAWARE (n = 44) and AWARE (n = 16) subgroups of FCPD subjects. The mean duration of diabetes was 9.3 ± 5.2 years in the UNAWARE group and 7.8 ± 4.7 years in the AWARE group, with glycemic control being moderate-to-poor in them. Both the UNAWARE (mean age 29.93 ± 8.2 years) and AWARE (30.8 ± 7.4 years) groups were relatively young and predominantly males. The mean BMI was low in both the groups (18.2 ± 6.5 kg/m$^2$ vs 18.5 ± 7.3 kg/m$^2$). Patients with FCPD in both the groups showed reduced stool elastase levels, suggestive of similar degrees of pancreatic exocrine insufficiency. All patients with FCPD received insulin as a basal bolus regimen. Other baseline parameters including mean insulin requirement, lipids and creatinine levels were comparable between both the groups, as demonstrated in **Table 1**: **Baseline** characteristics in patients with FCPD & hypoglycemia awareness/unawareness.

**Table 1. Algorithm of the study.**

| Demographic and clinical profile of patients | HU (n = 44) | HA (n = 16) | P value |
|---|---|---|---|
| Age (years) | 29.9 ± 8.2 | 30.8 ± 7.4 | 0.36 |
| Gender proportions (male: female) | 31:13 | 11:5 | 0.22 |
| BMI (Kg/m$^2$) | 18.2 ± 6.5 | 18.5 ± 7.3 | 0.44 |
| Mean duration of diabetes (years) | 9.3 ± 5.2 | 7.8 ± 4.7 | 0.19 |
| HbA1C (%) | 8.8 ± 2.2 | 9.2 ± 2.4 | 0.28 |
| Serum creatinine (mg/dl) | 1.2 ± 0.5 | 1.1 ± 0.4 | 0.15 |
| Serum LDL (mg/dl) | 116 ± 52.5 | 126 ± 62.8 | 0.29 |
| Serum Triglycerides (mg/dl) | 138 ± 65.4 | 129 ± 58.8 | 0.32 |
| Daily insulin requirement (U/day) | 29.2 ± 13.4 | 27.8 ± 12.6 | 0.62 |
| Fecal elastase (μg/g stool) | 40.8 ± 13.7 | 43.7 ± 12.3 | 0.53 |
| Patients on pancreatic enzyme supplementation (%) | 44 (100%) | 16 (100%) | - |
| **Frequency of hypoglycemia** | | | |
| Hypoglycemic episodes per week | 5.66 ± 2.60 | 3.83 ± 1.70 | **< 0.05**[*] |
| Nocturnal hypoglycemia episodes per week | 2.13±1.21 | 1.46 ± 0.82 | **< 0.05**[*] |
| Hypoglycemia unawareness episodes per week | 2.69 ± 1.33 | 1.03 ± 0.34 | **< 0.01**[*] |
| Hypoglycemia unawareness index | 0. 9± 0.4 | 0.07 ± 0.02 | **< 0.01**[*] |

BMI, body mass index; HbA1C, glycosylated hemoglobin; LDL-C, low density lipoprotein cholesterol

Values are presented as Mean ± SD, P < 0.05: Statistically significant.

Abbreviations: HU, Hypoglycemia Unaware group; HA, Hypoglycemia aware group;

## Assessment of hypoglycemia in patients with FCPD

The prevalence and patterns of hypoglycemia and HUA between the two groups of patients with FCPD were compared (**Table 1**). Analysis of quantitative data showed that 86% of patients in the UNAWARE group had at least one episode of hypoglycemia when compared to 43% patients in the AWARE group (*P* value: < 0.05). The UNAWARE group had significantly higher weekly episodes of hypoglycemia (p value: < 0.05), and nocturnal hypoglycemia (*P* < 0.02), than the AWARE group. Similarly, weekly occurrences of hypoglycemia unawareness (2.69 *vs*. 1.03 per week, *P* < 0.05) and the HUI (0.9 *vs*. 0.1, *P* < 0.01) were significantly higher in the UNAWARE group. Compared to the AWARE group, a substantially higher proportion of patients in the UNAWARE group had clinically significant hypoglycemia (*n* = 70 *vs*. 33, *P* < 0.05) as well as severe hypoglycemia (n = 2 *vs*. 0). Further, the patients in the UNAWARE group had spent 60.8% of the time in hypoglycemia, compared to 42.1% of time spent in hypoglycemia by the AWARE group (*P* < 0.05).

## Assessment of GV

Overall, the parameters of GV namely MAGE (183.6 ± 93.6mg/dl *vs*. 147.6 ± 79.3 mg/dl), CONGA (127.8 ± 19.8 mg/dl vs 115.2 ± 18.2 mg/dl) and SD (109.8 ± 25.2 *vs*. 86.4 ± 21.6 mg/dl) were significantly higher for the UNAWARE group when compared to the AWARE group (all; p < 0.05). Similarly, the parameters of % CV and MODD were significantly higher in the patients from UNAWARE group (*P* < 0.05), while the mean TIR was significantly lower (p < 0.05) for the UNAWARE group when compared to that in the AWARE group. The GV indices to characterize hyperglycemia were significantly higher in the AWARE group and the GV indices to characterize hypoglycemia were significantly higher in the UNAWARE group. The results of GV in the two groups are outlined in **Table 2**.

## Cardiac autonomic function tests

The comparison of Cardiac autonomic functions between the two groups of patients with FCPD with respect to the sympathetic and parasympathetic abnormalities is presented in **S1 Table**. Parasympathetic function assessment showed that variabilities in the E:I ratio and standing heart rate were significantly deranged in the UNAWARE group when compared to the AWARE group. Derangement in standing blood pressure was the most significant sympathetic abnormality seen in the UNAWARE group (59%) when compared to AWARE group (33%).

## Mixed meal challenge test

The results of MMCT were compared between the UNAWARE (n = 44) the AWARE (n = 16) subgroups of FCPD subjects and the healthy, normoglycaemic subjects (controls; n = 11). The mean values of glucose, C-peptide and glucagon levels at various time points in each group is given in **Table 3**.

The analysis of beta-cell secretory patterns following the MMCT revealed significantly greater excursions in both the UNAWARE and the AWARE groups when compared to the healthy, normoglycaemic subjects. Notably, C-peptide deconvolution studies demonstrated significantly lower C-peptide levels in both groups of patients with FCPD than in the controls (p <0.01), However, no significant differences in C- peptide levels were noted between the AWARE and UNAWARE groups (p = 0.18) as shown in **Fig 2A**.

In the AWARE group, the mean basal plasma glucagon level (142.03 ± 25.9 ng/ml) was significantly higher than the controls (92.34 ± 15.45 ng/ml, *P* < 0.05) and the UNAWARE group

**Table 2. Glycemic variability indices in patients with FCPD and hypoglycemia awareness and unawareness.**

| Indices of glycemic variability | Hypoglycemia Unaware group (n = 44) | Hypoglycemia aware group (n = 16) | P value |
|---|---|---|---|
| MAGE* (mg/dl) | 183.6 ± 93.6 | 147.6 ± 79.2 | < **0.05**[*] |
| CONGA-6* (mg/dl) | 127.8 ± 59.8 | 115.2 ± 48.6 | < **0.05**[*] |
| SD* (mg/dl) | 109.8 ± 25.2 | 86.4 ± 21.6 | < **0.05**[*] |
| MODD* (mg/dl) | 86.4 ± 34.2 | 55.8 ± 28.8 | < **0.05**[*] |
| CV* (%) | 40.2 ± 16.3 | 32.5 ± 11.5 | < **0.05**[*] |
| Mean value | 283.9 ± 129.6 | 265.6 ± 104.8 | 0.11 |
| TIR (Time in Range) (%) | 24 ± 19.6 | 36.7 ± 15.3 | < **0.05**[*] |
| **Indices of hypoglycemia** | | | |
| GRADE-hypo* (%) | 17.6 ± 8.7 | 9.2 ± 4. 4 | < **0.05**[*] |
| AUC < 70* (mg/dl) | 2. 8 ± 1.2 | 1.1 ± 0.7 | < **0.05**[*] |
| TSB* < 70 (mg/dl) | 11.6 ± 5. 4 | 5.9 ± 3.2 | < **0.05**[*] |
| LBGI* | 9.3±5.1 | 4.8 ± 2.6 | < **0.05**[*] |
| **Indices of hyperglycemia** | | | |
| GRADE-hyper* (%) | 72.5 ± 32.4 | 92.8 ± 29.8 | < **0.05**[*] |
| AUC >180* (mg/dl) | 29.3 ± 11.9 | 50.3± 24.8 | < **0.01**[*] |
| TSA >180* (mg/dl) | 48.2 ± 26.2 | 76.9± 30.5 | < **0.05**[*] |
| HBGI* | 12.1± 8.8 | 18.7± 10.3 | < **0.05**[*] |

Values are presented as Mean ± SD, *P* value < 0.05: Statistically significant

Abbreviations and its expansions

CONGA: Continuous overall net glycemic action.

% CV: coefficient of variance.

MAGE: mean amplitude of glycemic excursion.

MODD: absolute means of daily difference.

SD: standard deviation of the mean of the glucose values recorded on the sensor.

AUC: area under curve.

AUC > 180, AUC above 180mg/dL.

AUC < 70: AUC below 70mg/dL.

GRADE -hyper: glycemic risk assessment diabetes equation of hyperglycemia.

GRADE- hypo: glycemic risk assessment diabetes equation of hypoglycemia.

HBGI: High Blood Glucose Index. LBGI, Low Blood Glucose Index.

TSA > 180: time spent with blood glucose level above 180mg/dL

TSB < 70: time spent with blood glucose level below 70mg/dL

(43.62± 18.44ng/ml, $P < 0.01$). Similarly, the post prandial glucagon levels were significantly higher in the AWARE group when compared to the UNAWARE group (p < 0.01), and the controls (p: < 0.05) across all the time-points as shown in **Fig 2B**.

Correlation analysis for glucagon secretion (AUC-glucagon) in the UNAWARE group showed significantly negative correlations with the HUI ($r$ = -0.74, $P < 0.05$), nocturnal hypoglycemia episodes ($r$ = -0.69, $P < 0.05$) and GV indices namely MAGE ($r$ = -0.66, $P < 0.05$) Standard Deviation for the mean of glucose values recorded on the sensor ($r$ = -0.62, $P < 0.05$), GRADE-hypo ($r$ = -0.59, $P < 0.05$) and TSB < 70 ($r$ = -0.60, P: < 0.05). On the other hand, the glucagon secretion (AUC-glucagon) in the AWARE group showed significant positive correlations with two-hour post-prandial hyperglycemia ($r$ = 0.61, $P < 0.05$), GV indices of MAGE ($r$ = 0.62, $P < 0.05$) and GRADE-hyper ($r$ = 0.56, $P < 0.02$). On regression analysis, glucagon levels correlated significantly with higher HUI and MAGE in the UNAWARE group, while BMI along with glucagon levels correlated with higher MAGE in the AWARE group.

**Table 3. Glucose, C-peptide and glucagon levels in response to MMCT in patients with FCPD and hypoglycemia awareness / unawareness.**

| Time points (mins) | Glucose (mg/dl)* | | C-peptide (ng/dl)** | | Glucagon (pg/ml) | | P value |
|---|---|---|---|---|---|---|---|
| | Group 1# (n = 11) | Group 2# (n = 11) | Group 1# (n = 11) | Group 2# (n = 11) | Group 1# (n = 11) | Group 2# (n = 11) | |
| 0 | 117 ± 34 | 102.8 ± 45 | 0.56 ± 0.18 | 0.64 ± 0.22 | 142.0 ± 57 | 43.6 ± 14 | < **0.01** |
| 15 | 127 ± 40 | 131±61 | 0.7 ± 0.3 | 1.05 ± 0.46 | 156.5 ± 50 | 52.4 ± 24 | < **0.01** |
| 30 | 141 ± 56 | 143 ± 39 | 0.80 ± 0.41 | 1.49 ± 0.51 | 152.1 ± 44 | 59.4 ±20 | < **0.01** |
| 45 | 159 ± 68 | 150 ± 42 | 0.92 ± 0.52 | 1.71 ± 0.50 | 166.0 ± 68 | 64.8 ± 30 | < **0.01** |
| 60 | 168 ± 48 | 170 ± 82 | 0.99 ± 0.40 | 1.91 ± 0.78 | 165.2 ± 70 | 75.7 ± 28 | < **0.01** |
| 90 | 181 ± 83 | 178 ± 52 | 1.16 ± 0.45 | 1.92 ± 0.75 | 195.2 ± 81 | 84.3 ± 33 | < **0.01** |
| 120 | 182 ± 77 | 165 ± 70 | 1.29 ± 0.30 | 1.96 ± 0.64 | 198.7 ± 88 | 83.6 ± 37 | < **0.01** |
| 150 | 183 ± 61 | 148 ± 45 | 1.29 ± 0.33 | 1.82 ± 0.85 | 162.5 ± 31 | 64.5 ± 21 | < **0.01** |
| 180 | 180 ± 75 | 138 ± 66 | 1.26 ± 0.51 | 1.50 ± 0.46 | 138.1 ± 60 | 48.5 ± 22 | < **0.01** |

Values are presented as Mean ± SD, P value < 0.05: Statistically significant

*Glucose levels: P value for all time points: Not significant

**C-peptide levels: P value for all time points: Not significant.

Group 1: Patients with FCPD and hypoglycemia awareness

Group 2: Patients with FCPD and hypoglycemia unawareness

## Discussion

In this study, the baseline characteristics of the AWARE and UNAWARE groups of patients with FCPD were comparable. The mean age at diagnosis of FCPD was 30.8 years and 29.9 years in the AWARE and UNAWARE groups respectively with six patients (10%) diagnosed within 20 years of age, and 12 subjects (20%) diagnosed after 30 years of age. A previous study showed a similar mean age of diagnosis of FCPD [27]. Another study has reported a trend towards diagnosing the disease in the fourth or fifth decades of their life [3]. In this study, the majority of patients in either group were males (70%) with low BMI (<18.5kg/m²), the latter corroborating with the classical description of FCPD. About 5 patients with FCPD (10%) in the UNAWARE and 3 patients with FCPD (20%) in the AWARE group had BMI more than 24kg/m², as described in recent studies [3]. It is possible that better nutrition and control of

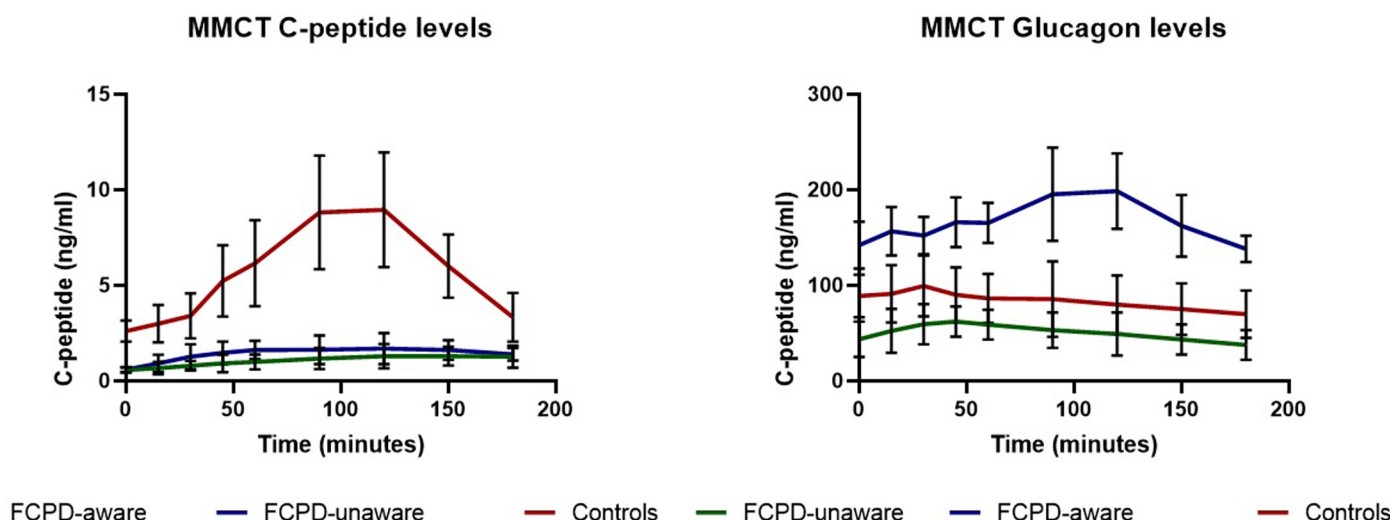

**Fig 2.** (**a**). C-peptide levels in patients with FCPD and controls during MMCT. (**b**). Glucagon levels in patients with FCPD and controls during MMCT.

steatorrhea with pancreatic enzyme supplementation and nutritional management in addition to timely glycemic control led to an improvement in BMI status.

Previous studies from India have shown poor glycemic control in patients with FCPD [27, 28]. In this study, the glycemic control of patients with FCPD ranged from moderate to poor, as evidenced by the HbA1C levels in both groups. This can be attributed to the duration of diabetes, pronounced beta cell destruction, nutritional mal-absorption, improper insulin dosage and administration by the patients. Furthermore, pancreatic exocrine insufficiency as documented by a lower fecal elastase level ($< 200$ μg/g of stool) was present in all the patients with FCPD and both groups received treatment with pancreatic enzyme supplements, thus ensuring that they were matched with respect to the degree of fat malabsorption.

The present study demonstrated that 75% (n = 45/60) of patients with FCPD developed hypoglycemia during the two-week follow-up period, as documented by an FGMS reading of glucose level less than 70 mg/dL. A previous study had shown that up to 44% of subjects of FCPD developed hypoglycemia when they were assessed with continuous glucose monitoring (CGM) [28]. This difference was probably due to a longer duration of glucose monitoring in our study (2 weeks of FGMS sensor-based glucose monitoring), whereas the previous study had performed only 3 days of CGM based glucose monitoring. A study conducted on patients who had undergone total pancreatectomy had shown a higher rate of hypoglycemia, with all patients (n = 56) experiencing symptomatic hypoglycemia and about 40% developing severe hypoglycemia [29].

On comparing the patterns of hypoglycemia amongst the two groups, the proportion of patients with hypoglycemia was nearly double in the UNAWARE group (88.36%) when compared to that of the AWARE group (43.75%), with a significantly higher HUI in the hypoglycemia UNAWARE group. This indicates that 90% of the hypoglycemic episodes in the UNAWARE group were unnoticed in this study. Despite the clinical importance of this feature, there is a paucity of similar studies evaluating hypoglycemia unawareness in patients with FCPD. A previous study on patients with T1DM and T2DM, showed variable ranges of hypoglycemia unawareness between 10 to 20% with higher numbers in T1DM [30]. As the pathophysiological mechanisms of T1DM are entirely different from FCPD, the comparison across the various subtypes of diabetes is difficult.

In this study, patients in the UNAWARE group had significantly higher numbers of weekly episodes of hypoglycemia, nocturnal hypoglycemia and hypoglycemia unawareness, with a prolonged duration of hypoglycemia when compared to that of the AWARE group. Awareness of hypoglycemia is mediated through a complex pathway involving the sympathoadrenal network and the counter-regulatory hormones, particularly glucagon [31]. Hypoglycemia Associated Autonomic Failure (HAAF) is mediated through a loss of insulin and glucagon response and an attenuated sympathoadrenal response, mediated through various postulated pathways such as the brain fuel hypothesis, brain metabolism hypothesis, and cerebral network hypothesis. Recurrent hypoglycemia itself reduces the sympatho-adrenal response leading to HAAF [32]. A study conducted by Cruckendall *et al.*, [33] had reported 7% mortality in patients with hypoglycemia (blood glucose level $< 70$mg/dl), signifying the importance of preventing hypoglycemia unawareness. Patients with FCPD and hypoglycemia unawareness are prone for prolonged episodes of nocturnal hypoglycemia, thereby predisposing them to cardiac dysfunction and sudden fatalities. Therefore, more vigorous blood glucose monitoring, and judicious use of insulin with a less stringent glycemic control should be the target to be achieved in patients with FCPD and hypoglycemia unawareness.

The evaluation for cardiac autonomic neuropathy in patients with FCPD revealed that 65% (39/60) of the participants had either sympathetic or parasympathetic autonomic dysfunction. These results were comparable to a previous study done at our institute, which had

demonstrated 63.3% cardiac autonomic dysfunction in patients with FCPD [23]. In this study, patients with FCPD and hypoglycemia unawareness demonstrated a significantly higher derangement of sympathetic (postural drop in BP) and parasympathetic (alteration in HR, E:I Ratio and standing HR) functions, which can partly explain the higher preponderance of severe hypoglycemia and hypoglycemia unawareness in them, as seen in a previous study [34].

On assessment of glycemic variability (GV) between the groups, the overall GV indices including MAGE, CONGA, SD, % CV and TIR showed more derangement in the UNAWARE group, suggesting that the latter had higher intra-day glucose fluctuations. While a previous CGM-based study from India in patients with FCPD suggested higher GV in them [28], the current study has differentiated GV responses amongst FCPD patients with or without hypo-glycemia unawareness. Further, the previous study was done using CGMS over three days in hospitalized patients, whereas in the current study, we used the 14-day record generated by FGMS monitoring, without interrupting the daily routine of patients with FCPD.

The evaluation of indices of hypoglycemia demonstrated poorer outcomes for the indices namely GRADE-hypo, AUC < 70, TSB < 70 and LBGI in the UNAWARE group. Contrast-ingly, the AWARE group had significantly higher derangements for the GV measures of hyperglycemia namely GRADE-hyper, AUC >180, TSB >180 and HBGI. Our findings suggest that patients with FCPD and hypoglycemia unawareness had higher glycemic variability due to hypoglycemic episodes, with GV-hypo parameters being worse when compared to a previ-ously reported CGM-study [28]. On the contrary, glycemic variability seen in patients with FCPD and hypoglycemia awareness was largely contributed by hyperglycemic episodes, thus predisposing them to risk for future hyperglycemic crisis. Correlation analysis between the gly-cemic variability parameters and patient characteristics showed a significant negative correla-tion between BMI and MAGE in both the groups, with BMI demonstrating a negative correlation with CONGA in patients with FCPD and hypoglycemia unawareness. A previous study has demonstrated that a lower BMI is associated with a higher GV [35], implying that FCPD patients with significant weight loss have a greater propensity for glycemic fluctuations and developing diabetic complications [18]

On analyzing the data from MMCT performed in subsets of patients with FCPD & hypogly-cemia unawareness and awareness and in controls, the basal and stimulated C- peptide levels in both the FCPD groups were observed to be significantly lower than that of the controls, indicating the presence of severe β- cell loss in them. Fasting and meal-stimulated glucagon values showed contrasting findings in the AWARE and UNAWARE groups. In the AWARE group, both fasting and stimulated glucagon levels were significantly higher than the controls, with peak values noted at a 120-minute post-meal challenge. In contrast, the UNAWARE group, had significantly reduced fasting and post-meal glucagon levels compared to controls. A correlation analysis of glucagon levels (0–180 minutes) following mixed-meal challenge also showed divergent findings in the two groups. In patients with FCPD and hypoglycemia aware-ness, glucagon levels correlated positively with the post meal blood glucose levels ($r = 0.70$, $P < 0.01$) and HUI, suggesting that hyperglucagonemia in this group increased the risk of postprandial hyperglycemia while protecting such patients from hypoglycemia unawareness. A positive correlation of glucagon levels with MAGE and GV-hyperglycemia variables in patients with FCPD and hypoglycemia awareness, further suggested that increased glucagon levels may play a key role in modulating hyperglycemia associated GV in this group. On the other hand, significant negative correlations were observed between glucagon levels and HUI ($r = -0.74$, $P < 0.01$) and nocturnal hypoglycemia ($r = -0.69$, $P < 0.01$) in patients with FCPD and unaware of hypoglycemia, coupled with significant negative correlations of glucagon levels with the SD, MAGE and GV-hypoglycemia indices in this group. These findings suggest that reduced glucagon levels increased the risk of hypoglycemia unawareness and hypoglycemia-

mediated glycemic variability in this group. The correlations of glucagon levels with HUI and GV indices in both groups were maintained after adjustment for cardiac autonomic dysfunction, suggesting that glucagon levels independently influenced hypoglycemia unawareness and GV parameters in FCPD.

The neuro-endocrine regulation of pancreatic glucagon is complex and mediated by indirect signaling of alpha cells [36] the delta cell secretory products [37], the autonomic nervous system [38], the gut incretins [39] and autocrine signals [40]. During hypoglycemia, a decrease in somatostatin secretion from the delta cells occurs and this decrement is amplified by factors released from the alpha and beta cells [41]. This could also stimulate higher glucagon secretion in patients with diabetes leading to symptomatic hypoglycemia. This feature may vary based on the duration of the disease in such patients. Studies in animal models have demonstrated that mice lacking the insulin receptor on alpha cells ($\alpha$ IRKO) feature hyperglycemia and hyperglycemia [42]. Alpha cell insulin resistance could be a pivotal factor in causing hyperglucagonemia. It needs to be investigated thorough prospective studies if this could be a possible glucoregulatory feature in patients with FCPD and hypoglycemia awareness.

Glucagon levels in patients with chronic pancreatitis has been researched earlier. A study by Knop *et al.*, compared patients with chronic pancreatitis, impaired glucose tolerance (IGT) and diabetes with normoglycaemic individuals in response to oral and intravenous glucose tolerance tests. This study reported impaired suppression of glucagon secretion in patients with chronic pancreatitis and IGT or diabetes in response to an oral glucose tolerance test [43]. In the study by Lund *et al.*, an elevated glucagon response was observed at baseline and in response to oral glucose tolerance test, whereas a suppressed glucagon response was noted to an intravenous glucose tolerance test in such patients [44]. Furthermore, a study conducted by Yajnik *et al.*,[16] had reported elevated glucagon levels in patients with FCPD following an oral glucose tolerance test, whereas a study by Mohan *et al.*, had shown preserved fasting glucagon levels in patients with FCPD [45].

The hypothesis of an extra-pancreatic source of glucagon had been evidenced earlier in patients who had undergone total pancreatomy. A study in patients who had undergone partial pancreatomy reported elevated glucagon levels in response to an oral glucose tolerance test [46]. Studies in animal models have also demonstrated elevated glucagon levels in response to arginine stimulation, following total pancreatomy, possibly due to extra pancreatic source of glucagon [47]. Though there is no previous data elucidating a correlation between glucagon levels and GV in patients with FCPD, previous studies conducted in patients with type 1 diabetes mellitus suggest that glucagon levels have a significant influence on post-prandial hyperglycemia [48] and an independent negative correlation with GV. These findings are comparable to our study though the mechanisms in FCPD may be different. Notably, the fluctuations in glucagon levels influencing hypoglycemia awareness and GV in such patients may be modulated by glucoregulatory peptides namely glucagon like peptide -1 (GLP-1), gastric inhibitory peptide-1 (GIP 1) [49] which needs further studies.

Hypoglycemia awareness and glycemic variability in patients with FCPD is maintained through a constant interplay between autonomic nervous system and the glucagon axis, with glucagon levels showing an independent and dichotomous relation with these parameters [50]. Relatively reduced glucagon levels contribute to hypoglycemia unawareness, prolonged hypoglycemia and increased glycaemic variability whereas, hyperglucagonemia provides better hypoglycemia awareness at the cost of postprandial hyperglycemia and increased glycaemic variability. The exact causative factor of this feature is unclear. Further research is essential to evaluate the pathophysiological mechanisms explaining this heterogeneity in glucagon levels and possible source of elevated glucagon levels, despite extensive pancreatic destruction in patients with FCPD.

The current study is novel to evaluate the complex interrelationship between glucagon levels, glycemic variability and hypoglycemia unawareness in a cohort of patients with FCPD. We have employed rigorous 14-day AGP recordings using CGMS with concomitant 6-point-SMBG profile using a glucometer. However, utilizing stepped hypoglycemic clamps would have been the ideal technique for assessing hypoglycemia unawareness in patients with FCPD, but that remains beyond the scope of the current study. Further, the assessment of levels of counter-regulatory hormones during hypoglycemia would have provided a comprehensive view of hypoglycemia unawareness in our study population. Nevertheless, the observations reported in this study deem merit for further research through prospective studies in patients with FCPD.

## Supporting information

**S1 Table. Cardiac autonomic function assessment between groups.**
(DOCX)

**S1 File. Clarke's questionnaire.**
(DOCX)

**S2 File. Cardiac autonomic function test.**
(DOCX)

**S3 File. Glycemic variability indices and their expansions.**
(DOCX)

## Acknowledgments

The authors thank the participants for their involvement in this study.

## Author Contributions

**Conceptualization:** Riddhi Dasgupta, Nihal Thomas.

**Data curation:** Riddhi Dasgupta, Santhya Seenivasan, K. U. Lijesh.

**Formal analysis:** K. U. Lijesh.

**Funding acquisition:** Riddhi Dasgupta, Santhya Seenivasan.

**Investigation:** Riddhi Dasgupta, Santhya Seenivasan, Gracy Varghese, Pamela Christudoss.

**Methodology:** Shajith Anoop, Santhya Seenivasan, Flory Christina.

**Project administration:** Riddhi Dasgupta, Shajith Anoop, Deepu David, Sudipta Dhar Chowdhury, Nihal Thomas.

**Resources:** Riddhi Dasgupta, Deepu David, Sudipta Dhar Chowdhury, Thomas V. Paul, Nihal Thomas.

**Supervision:** Riddhi Dasgupta, Felix K. Jebasingh, Shajith Anoop, Nihal Thomas.

**Validation:** Gracy Varghese, Pamela Christudoss, Nihal Thomas.

**Writing – original draft:** Felix K. Jebasingh, K. U. Lijesh.

**Writing – review & editing:** Riddhi Dasgupta, Felix K. Jebasingh, Shajith Anoop, Mathews Edatharayil Kurian, Thomas V. Paul, Nihal Thomas.

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
