## [Decision Letter · Decision Letter 0]

11 May 2022

PONE-D-21-10177Comprehensive Evaluation of patterns of Hypoglycemia Unawareness (HUA) and Glycemic Variability (GV) in patients with Fibro Calculous Pancreatic Diabetes (FCPD):  A cross-sectional study from South IndiaPLOS ONE

Dear Dr. Thomas,

Thank you for submitting your manuscript to PLOS ONE. After careful consideration, we feel that it has merit but does not fully meet PLOS ONE’s publication criteria as it currently stands. Therefore, we invite you to submit a revised version of the manuscript that addresses the points raised during the review process.

We look forward to receiving your revised manuscript.

Kind regards,

Mohamed A Elrayess

Academic Editor

PLOS ONE

**Journal requirements:**

2. Thank you for including your ethics statement:  "This cross-sectional study was approved by the institutional review board (IRB) for ethics in research on humans (IRB number 10788 dated 01 August 2017).The study adhered to guidelines of the declaration of Helsinki, 2013. ".  

3. Please include your actual numerical p-values in Table S1.

“The authors thank the participants for their involvement in this study. This study was supported by a research grant from the Research Society for Study of Diabetes in India (RSSDI) ; Grant No: RSSDI/HQ/GRANTS/ 2018/460. The funding body had no role in study design, data collection and analysis,  preparation, review of the manuscript and decision to publish.”

“This study was supported by a research grant awarded to SS from the Research Society for Study of Diabetes in India (RSSDI) ; Grant No: RSSDI/HQ/GRANTS/ 2018/460. The funding body had no role in study design, data collection and analysis,  preparation, review of the manuscript and decision to publish.”

**Additional Editor Comments:**

Dear Dr Thomas,

In this manuscript, the authors examined the association of hypoglycemia unawareness and glycemic variability with glucagon secretion and cardia autonomic function in patients suffering from Fibrocalculous pancreatic diabetes (FCBD). The reviewers recommended that you make minor amendments to your manuscript. Please respond within the next 14 days to all comments raised by the reviewers.

Reviewers' comments:

Reviewer's Responses to Questions

**Comments to the Author**

1. Is the manuscript technically sound, and do the data support the conclusions?

Reviewer #1: Yes

Reviewer #2: Yes

2. Has the statistical analysis been performed appropriately and rigorously? 

Reviewer #1: I Don't Know

Reviewer #2: Yes

3. Have the authors made all data underlying the findings in their manuscript fully available?

Reviewer #1: Yes

Reviewer #2: Yes

4. Is the manuscript presented in an intelligible fashion and written in standard English?

Reviewer #1: Yes

Reviewer #2: Yes

5. Review Comments to the Author

Reviewer #1: This is an intersting research article investigating predominantly the effect of hypoglycemia unawareness on glycemic variability on FCPD. The article is well written and provides useful additional information to the field. A significant number of patients have been recruited and the data is clear.

I have minor corrections:

1. Abstract: Please provide at least one introductory sentence in the background.

2. Please state/confirm whether the participants are taking any other class of medications e.g. statins.

3. Figure 2 is blurry and difficult to see. I'm not sure if this is an upload issue but it should be sorted.

4. Check grammar throughout, including missing full stops.

Reviewer #2: In this manuscript, the authors examined the association of hypoglycemia unawareness and glycemic variability with glucagon secretion and cardia autonomic function in patients suffering from Fibrocalculous pancreatic diabetes (FCBD). Reduced fasting and postprandial glucagon levels were observed to be significantly associated with the unaware group and were negatively correlated with hypoglycemia and cardia autonomic functions. This is an interesting article and will be useful for diabetes researchers and clinicians as it suggests a link between low levels of glucagon and hypoglycemia unawareness in FCBD. I recommend it for publication with minor revision:

Please discuss a little more on the significance of increased glucagon levels in the aware group.

I also spotted some typos that are listed below.

1. Fibrocalculous instead of fibrocalculous.

2. Line 66 Up to instead of upto

3. Line 68, most common instead of commonest.

6. PLOS authors have the option to publish the peer review history of their article (what does this mean?). If published, this will include your full peer review and any attached files.

Reviewer #1: No

Reviewer #2: No

---

## [Author Response · Author response to Decision Letter 0]

27 May 2022

RESPONSES TO QUERIES OF EDITOR & REVIEWERS

QUERIES OF ACADEMIC EDITOR

Q1: Please amend your current ethics statement to include the full name of the ethics committee/institutional review board(s) that approved your specific study. Once you have amended this/these statement(s) in the Methods section of the manuscript, please add the same text to the “Ethics Statement” field of the submission form (via “Edit Submission”).

Authors’ reply: Thank you for the suggestion. In the revised draft of the manuscript, we have stated as “This cross-sectional study was approved by the institutional review board (IRB) for ethics in research on humans (IRB number 10788 dated 01 August 2017) of Christian Medical College (CMC) Vellore, India” in lines 78 to 80 of the revised draft. This has also been added in the section on ethics statement of the submission form. 

Q2: Please include your actual numerical p-values in Table S1.

Authors’ reply: Thank you for the comment. We have mentioned the exact p values for the variables in supplementary table S1. 

Q3: We note that you have provided funding information that is not currently declared in your Funding Statement. However, funding information should not appear in the Acknowledgments section or other areas of your manuscript. We will only publish funding information present in the Funding Statement section of the online submission form.

Authors’ reply: Thank you for the comment. We have deleted the section on funding information in the acknowledgment of the manuscript and have mentioned the same in the funding statement of the online submission form. 

Q4: Please include your amended statements within your cover letter. We will change the online submission form on your behalf.

Authors’ reply: Thank you for the kind gesture. We have mentioned the funding disclosure in the cover letter as “Disclosure: This study was supported by a research grant from the Research Society for Study of Diabetes in India (RSSDI); Grant No: RSSDI/HQ/GRANTS/ 2018/460. The funding body had no role in study design, data collection and analysis, preparation, review of the manuscript and the decision to publish.

Q5: Please review your reference list to ensure that it is complete and correct. If you have cited papers that have been retracted, please include the rationale for doing so in the manuscript text, or remove these references and replace them with relevant current references. Any changes to the reference list should be mentioned in the rebuttal letter that accompanies your revised manuscript. If you need to cite a retracted article, indicate the article’s retracted status in the References list and also include a citation and full reference for the retraction notice.

Authors’ reply: Thank you for the instruction. We cross checked the references on Pubmed and have added extra references (ref 36 to 41) in the section on discussion. No retracted references have been cited. All references confirm to the format stipulated by PLoS One. 

QUERIES OF REVIEWER 1

Q1: Abstract: Please provide at least one introductory sentence in the background. 

Authors’ reply: Thank you for the valuable comment. We have revised the abstract in the revised version. It now reads as “Background: Hypoglycemia unawareness (HUA) in patients with FCPD is common with an unclear etiology. We evaluated the prevalence, characteristics of HUA, glycemic variability (GV), its possible association with pancreatic glucagon secretion & cardiac autonomic function in patients with FCPD.

Q2. Please state/confirm whether the participants are taking any other class of medications. e.g., statins.

Authors’ reply: This is an important comment. In this study, the patients with FCPD were not on statins or beta blockers or any other medication that could interfere with hypoglycemia awareness. This has now been mentioned in the section on exclusion criteria appearing on lines 95 to 101 of the revised manuscript. 

Q3. Figure 2 is blurry and difficult to see. I'm not sure if this is an upload issue but it should be sorted.

Authors’ reply: Thank you for notifying this error. We have configured Fig 2(a) & (b) to 381 KB for optimal resolution. 

Q4. Check grammar throughout, including missing full stops.

Authors’ reply: Thank you for the keen observation. We have checked the manuscript and rectified the punctation and grammatic errors in the text, tables and captions. 

QUERIES OF REVIEWER 2

Q1: Please discuss a little more on the significance of increased glucagon levels in the aware group.

Author’s reply: We thank you for the valuable suggestion. In the section on discussion (lines 411 to 419), we have included a paragraph on the possible etiological factors regulating glucagon levels in the aware group. 

Q2: I also spotted some typos that are listed below. 

a) Fibrocalculous instead of fibrocalculous.

b) Line 66 Up to instead of upto.

c) Line 68, most common instead of commonest

Author’s reply: Thank you for the observations. In the revised draft, we have rectified typographical and punctuation errors throughout the manuscript including tables and captions.

---

## [Editor Report · Decision Letter 1]

21 Jun 2022

Comprehensive Evaluation of patterns of Hypoglycemia Unawareness (HUA) and Glycemic Variability (GV) in patients with Fibro Calculous Pancreatic Diabetes (FCPD):  A cross-sectional study from South India

PONE-D-21-10177R1

Dear Dr. Thomas,

We’re pleased to inform you that your manuscript has been judged scientifically suitable for publication and will be formally accepted for publication once it meets all outstanding technical requirements.

Kind regards,

Mohamed A Elrayess

Academic Editor

PLOS ONE
---

## [Editor Report · Acceptance letter]

27 Jun 2022

PONE-D-21-10177R1 

Comprehensive Evaluation of patterns of Hypoglycemia Unawareness (HUA) and Glycemic Variability (GV) in patients with Fibrocalculous Pancreatic Diabetes (FCPD):  A cross-sectional study from South India 

Dear Dr. Thomas:

I'm pleased to inform you that your manuscript has been deemed suitable for publication in PLOS ONE. Congratulations! Your manuscript is now with our production department. 

Kind regards, 

on behalf of

Dr. Mohamed A Elrayess 

Academic Editor

PLOS ONE